# OpenReview forum: "Cultivating Pluralism In Algorithmic Monoculture: The Community Alignment Dataset"
_ICLR.cc/2026/Conference — ICLR 2026 Poster_

### Official Review · Reviewer_rAhP · 2025-10-26

**Soundness:** 3
**Presentation:** 3
**Contribution:** 3
**Rating:** 6
**Confidence:** 4

**Summary:**

This work 1) identifies that general-purpose LMs are in the same "monoculture", that they struggle to represent diverse dimensions of preferences; 2) collects a dataset called community alignment, with negative-correlated sampling and human annotations.

**Strengths:**

+ community alignment could be a useful dataset
+ the discussion on monoculture is interesting

**Weaknesses:**

- In the monoculture analysis part, all models considered (for example, in Figure 1) are aligned language models. There is increasing recent research on how aligned models suffer from generation diversity issues, while base models are better. [1-2] Would it help alleviate the monoculture problem if including base models in text generation?

- Aside from that, the study on monocluture also only uses general-purpose LMs. This kind of reduces the surprise of the results, as most of the major industry models are trained on highly overlapping internet texts and post-training recipes, while "culture, value" was never the most important consideration in their training process, so there is naturally very little variation. I wonder if there are any models specialized to the culture/value space from the past few years of related research that might break from this pattern.

- The "negative-correlated sampling", albeit its fancy name, is a prompt that asks for "diverse values". The authors hope that by asking for multiple "diverse" items in the prompt, the generated items would be different from each other. Well, empirically this might have worked with the selected model. I'm a bit unsure if this "negative-correlated" is very reliable: it seems like it is subject to the model's specific niche capability that might change from model to model, rather than a sweeping and general method.

- I appreciate the community alignment dataset and releasing it. I wonder in addition to analyzing its statistics in Sec 4.1, is there any way to prove its usefulness in actual model training/development? Like, somehow adapting an existing model with some subset of the data leads to improvement on something? It's unclear the utility of this resource at the moment; maybe it was described in the very long appendix.

[1] Zhang, Jiayi, et al. "Verbalized Sampling: How to Mitigate Mode Collapse and Unlock LLM Diversity."

[2] Karan, Aayush, et al. "Reasoning with Sampling: Your Base Model is Smarter Than You Think."

**Questions:**

please see above

---

> ### Author Response · Authors · 2025-11-21
> **Author Response**
>
> Thank you for your review and highlighting the potential utility of Community Alignment and the discussion on monoculture as strengths of the work.
>
> Addressing your questions below (abridged for space):
>
> 1. [In the monoculture analysis, all models are aligned language models. Would it help to use base models in text generation?]
>
> - Great question. We agree with the reviewer that there could be other ways of inducing diversity beyond NC sampling from chat models. However, since base models are not fine-tuned for chat interactions, they would necessitate customized sampling or prompt-engineering with examples to maintain chat quality. Since the goal of our dataset was specifically to measure subjective and heterogeneous preferences for chat settings, the use of base models was less appealing.
>
> 2. [The study on monoculture only uses general-purpose LMs. This reduces the surprise of the results, as most models are trained on highly overlapping internet texts and post-training recipes, "culture, value" was never the most important consideration in their training process. I wonder if there are any models specialized to the culture/value space that might break from this pattern.]
>
> - Good question; even if it is true that “"culture, value" was never the most important consideration in their training process,” algorithmic monoculture is becoming an increasingly important concern as these models are being deployed to a global user base. To your question about models that break from this pattern, Figure C.2 may offer some unexpected insights. For English, French, Italian, and Portuguese, all evaluated models present an algorithmic monoculture and align predominantly with the secular-rational / self-expression quadrant. However, in Hindi, Claude, Qwen, Llama, and Mixtral exhibit greater variation in values, whereas GPT and Gemini continue to stay strictly within the secular-rational / self-expression quadrant (they are also consistently the most secular-rational / self-expression-oriented across all 5 languages). This suggests that, even among major industry models, there are differences in how models adapt or not to different cultural and linguistic contexts.
>
> 3. [NC sampling is a prompt that asks for "diverse values". Empirically this might have worked with the selected model. I'm a bit unsure if it is very reliable: it seems like it is subject to the model's specific niche capability that might change from model to model, rather than a sweeping and general method.]
>
> - Figure 2 shows that NC sampling results in Pareto improvement of IW values coverage across a wide array of models, even though the prompt is generic enough to not even mention the IW values themselves. The result is especially striking given the same instruction was provided to all models without any prompt engineering for any specific model. Table F.1 shows qualitative examples of generations from these models that also supports this claim.
>
> 4. [I appreciate the community alignment dataset and releasing it. I wonder in addition to analyzing its statistics in Sec 4.1, is there any way to prove its usefulness in actual model training/development? Like, somehow adapting an existing model with some subset of the data leads to improvement on something? It's unclear the utility of this resource at the moment; maybe it was described in the very long appendix.]
>
> - To clarify, the four bolded points in Sec 4.1 are not simply dataset statistics—they highlight distinct use cases enabled by the Community Alignment (CA) dataset. For instance, the “prompt-level overlap in annotators” is a unique aspect of CA that facilitates new research directions in social choice theory for alignment. We have already spoken with several social choice experts who are leveraging CA as a testbed for exploring various preference aggregation methods for LLM alignment.
>
> - As another example, the “multilingual data” and representativeness of CA enables new research into alignment across heterogeneous preferences and cultures. One follow-up work has already used CA to identify interpretable response features that vary subjectively across demographic groups—such as whether a response “does not emphasize technology-based solutions” or “emphasizes environmental sustainability and eco-friendly options”—and has used such features for interpretable personalization. (We redact the citation to this work to preserve double blind anonymity, since the paper cites our de-anonymized paper)
>
> - We also note that Section 3 shows that the NC sampling methodology used to construct the CA dataset improves the ability for a diverse range of methods–prompt-steering, SFT, DPO, and GRPO–to learn preferences along the Inglehart-Welzel dimensions.
>
> Thank you again for your thoughtful review. Please let us know whether these comments address your concerns or whether there are still outstanding ones; we'd be happy to address any additional questions you may have.

---

> > ### Comment · Reviewer_rAhP · 2025-11-21
> >
> > I would like to thank the authors for the detailed response.

---

### Official Review · Reviewer_3SDc · 2025-10-27

**Soundness:** 4
**Presentation:** 3
**Contribution:** 4
**Rating:** 10
**Confidence:** 4

**Summary:**

The paper introduces a large-scale dataset for heterogenous preferneces, with 200k preference demonstrations from 15k humans across five countries, and multiple annotations per prompt/response combination for a subset of the dataset. Additionally, the paper demonstrates how the current method of sampling many responses from an instruction-tuned model may be insufficient for getting coverage over the breadth of human preferences, and introduce negatively correlated sampling to (at least partially) mitigate this.

**Strengths:**

S1: I quite like the insight that people's preferences vary from one another far more than model responses from leading LLMs do - an important selection bias to document and mitigate.
S2: The contributed dataset is massive (200k comparisons from annotators from five countries), and is a very valuable resource for the community, especially in future pluralistic reward modeling.
S3: Negatively correlated sampling could be a useful technique for future work for getting more diverse candidate responses.
S4: Prompt-level overlap (L460) is a particularly exciting feature of CA for studying variation in human preferences, and was not explored in previous work.
S5: "As of today, Community Alignment is the largest open-source multilingual preference dataset and the first to feature prompt-level overlap in annotators along with natural language explanations for choices." (L476-477). These are exciting features!

**Weaknesses:**

W1: While an understandable omission for the space constraints, there is very limited analysis on the actual generated preferences. E.g., how much do people actually disagree in practice (interannotator agreement rates)? What were some of the features of the natural language preference explanations provided? What kinds of topics were covered in the prompts? A potential future camera-ready version of the paper could benefit from some additional of the above analyses (but I do not consider them essential).

**Questions:**

Q1: For preference alignment dataset construction, the paper says that "we start with a base language model, sample responses from it, ..." (L85). I wanted to ask - is it truly a base model? As base models often exhibit low instruction-following skills / chat persona, I believe it is common to do SFT first on chat demonstrations, or few-shot chat elicitation (eg., https://arxiv.org/abs/2312.01552). Could the authors clarify exactly what they mean here?
Q2: L109: "producing respnoses in English that align with only 41% of human preferences". I had a hard time contextualizing this number - what exactly is meant here? What would a baseline be?
Q3: Was there any attempt to see if none of the responses represented the raters well, as with the apple/banana fruit example? If not, do the authors think that future work will need to try to address a coverage gap in preferences by eliciting this from raters, or would you expect that something like negatively-correlated sampling be sufficient for eliciting broad coverage?

---

> ### Author Response · Authors · 2025-11-21
> **Author response**
>
> Thank you for your review and appreciating the strengths of our paper!
>
> Addressing your questions (abridged for space):
>
> W1. [analysis on the generated preferences]
> - Thank you for this suggestion of additional analysis on the dataset itself. Though the CA dataset was released only recently, subsequent work has already provided greater insights on its contents. One recent study uses SAEs to identify how the candidate responses differ. They analyze 7 open-source datasets, including CA, and reveal that CA is the only dataset containing responses that vary along values (e.g. environmental sustainability, personal well-being and mindfulness), as opposed to primarily in style (e.g. Markdown) or refusal vs non-refusal.  Additionally, the study demonstrates that preferences for certain features in CA vary across demographic groups (e.g. country, gender, education level, etc). For example, the preference for the feature “offers cultural or spiritual reflections instead of concrete, practical details” varies across political groups. (We redact the citation to this work to preserve double blind anonymity, since the paper cites our de-anonymized paper)
>
> - With regards to your specific questions:
> The inter-annotator agreement rate, measured by Krippendorf’s $\alpha$, is 0.180 for the pre-specified prompts with at least 10 annotators each, indicating significant heterogeneity in preferences.
> All topics for pre-specified prompts are listed in Sec E.1 in the paragraph with heading “prompts”.
> The human-written prompt topics range from religion and philosophy to career and travel, etc.; the explanations focus on the overall conversation and the individual's reasoning / values behind their choice.
>
> Q1. ["we start with a base language model, sample responses from it, ..." Could the authors clarify?]
> - Good question; by base, we meant an existing language model upon which one might wish to perform additional personalization / pluralistic alignment, not a language model before SFT for chat. We’ve removed the potentially confusing use of “base” in the introduction.
>
> Q2: ["producing responses in English that align with only 41% of human preferences". what exactly is meant here? What would a baseline be?]
>
> - Great question; this number is computed by measuring the proportion of individuals lying outside the area covered by the LLM responses on the IW coordinate map (Figure 1). Specifically, we identify the minimum and maximum x and y coordinates among all 420 points representing LLM results, then calculate the percentage of human responses that lie within this defined range. Note that this is a very conservative measure of algorithmic monoculture as it only requires one model to be in the bottom left quadrant (survival / traditional quadrant) for those humans to be considered “covered”. A good baseline would be the percentage computed if the 420 LLM instances on the RHS of Figure 1 were sampled from a distribution matching that of the human population (LHS of Figure 1). For the US, when we compute this percentage using 10,000 bootstrap samples of n=420 drawn from the human population, we consistently find that 100% of humans are covered. In contrast, the actual LLM responses only cover 41%, which is a striking difference.
>
> Q3: [Any attempt to see if none of the responses represented the raters well? Do the authors think that future work will need to address a coverage gap by eliciting this from raters, or would something like negatively-correlated sampling be sufficient?]
>
> - Fantastic question; while negatively-correlated sampling substantially improves coverage, it is possible that there still exist coverage gaps. We did not explicitly look for coverage gaps in the Community Alignment data collection process, but we did so in a preliminary participatory study described in C.1 that preceded the large-scale human study in Sec 2.1. In this initial study, participants that were dissatisfied with all candidate responses were asked to write an alternate response. Additionally, every participant was given 4 additional prompts that they always wrote their own response to. All human-written responses were then qualitatively coded to identify what value considerations they reflected, and whether they aligned with the 4 dimensions considered for the preliminary study (which include two beyond the IW dimensions considered in the main text). Through this qualitative coding, we did not find any new salient dimensions beyond the four that we had pre-specified. Moreover, we found that individual preferences varied the most along the two IW dimensions, which motivated our subsequent focus on them.
>
> For future work, we are indeed conducting a larger-scale dataset collection in which participants write their own responses. We hope to directly compare the coverage of human-written responses to that of different candidate sampling strategies including NC sampling.
>
> Thank you again for your review!

---

> > ### Comment · Reviewer_3SDc · 2025-11-22
> >
> > Thank you so much for answering my questions! I appreciate the clarifications and minor paper edits, and am happy to keep my score.

---

### Official Review · Reviewer_Y7NQ · 2025-11-01

**Soundness:** 2
**Presentation:** 2
**Contribution:** 3
**Rating:** 4
**Confidence:** 4

**Summary:**

This paper examines how LLMs can address diverse and conflicting human preferences across cultural and political contexts. A large-scale multilingual study (N=15,000) shows human preferences vary more than LLM outputs. The authors introduce negatively-correlated candidate sampling to better capture this diversity and release Community Alignment, a large multilingual dataset (~200,000 comparisons) designed to improve LLM alignment with global populations.

**Strengths:**

1. This paper introduces a new perspective for evaluating preference dataset diversity using Inglehart and Welzel (IW) dimensions, revealing that 21 LLMs exhibit an “algorithmic monoculture” and are poorly aligned with human preferences.
2. The authors propose negatively-correlated sampling, an efficient method to enhance the diversity of LLM-generated responses.
3. They also release Community Alignment, an open-source multilingual preference dataset built with negatively-correlated sampling, containing about 200,000 comparisons from annotators across five countries, with samples balanced by age, gender, and ethnicity in three of them.

**Weaknesses:**

1. The paper lacks a related work section, making it difficult for readers to understand its position within existing research. For instance, it is unclear whether the proposed negatively-correlated sampling method is novel or adapted from prior work.
2. I am concerned about using only four dimensions—secular-rational vs. traditional and self-expression vs. survival values—to measure preference diversity. How representative are these dimensions overall? The authors should elaborate on their rationale and limitations.
3. The paper provides no quality analysis of the collected dataset, and its description is too brief. As a key contribution, the dataset’s quality and utility should be discussed in greater depth. It is also unclear why the authors did not train or evaluate models using their collected dataset.

**Questions:**

1. L179-180: we generate and curate three model responses to vary along one of four known dimensions
of variation in individual values. How do you check the quality of generated responses here?
2. Table 1 takes up substantial space but provides limited information, could the authors present these results in a more concise or effective format?
3. Why did the authors not train or evaluate models using their collected dataset, given that it is one of the key contributions of the paper?

---

> ### Author Response · Authors · 2025-11-21
> **Author response**
>
> Thank you for your review, and recognizing the trifecta of contributions from this paper, (algorithmic monoculture, NC sampling, Community Alignment).
>
> Responding to your concerns (abridged for space):
> 1. [The paper lacks a related work section. Is negatively-correlated sampling method novel or adapted from prior work]
> - Due to space limitations in the initial submission, we put our related work (2+ pages) to the appendix (Section B). In the updated submission, we have added an abridged related work section in the main text.
> - To clarify, NC sampling for preference dataset collection is a novel contribution of this work, not adopted from prior work (though the prompt is simple enough we do not claim that no one has ever used such a prompt for other purposes).
> 2. [I am concerned about using only four dimensions to measure preference diversity. How representative are these dimensions overall? The authors should elaborate on their rationale and limitations.]
> - Good question; the IW dimensions are derived from the most comprehensive longitudinal survey of values worldwide (World Values Survey, over 100 societies over multiple decades) and explain over 70% of the cross-national variance in the factor analysis conducted in the original study [1](https://www.worldvaluessurvey.org/wvs.jsp). Moreover, we additionally conducted an initial exploratory participatory study which confirmed the salience of these specific dimensions for capturing participant preferences for LLM responses (Appendix C.1).
> 3. [The paper provides no quality analysis of the collected dataset, and its description is too brief.]
> - We would like to point the reviewer to
> 	- Appendix E.2 which outlines the extensive quality checks performed on the dataset, including a training phase for all annotators requiring attention tests and open-ended justifications of their choices as well as audits throughout the data collection process;
> 	- Table 2 which highlights that CA substantially surpasses other open-source preference datasets in terms of demographic representativeness (and we have improved even further upon representativeness, as highlighted in the general response to all reviewers);
> 	- Section 4.1 (Novel Aspects of Community Alignment) as well as the Future Work paragraph in Appendix F which outlines the many new analyses and research that this dataset unlocks. To name a few:
> 		- Testing social choice and distributional alignment methods (due to prompt-level overlap in annotators);
> 		- Advancing alignment for heterogeneous populations and cultures  (as the most demographically representative and multilingual preference dataset to date);
> 		- Advancing personalization research (due to high # of conversations per annotator); One subsequent work has already used CA to develop an interpretable personalization method (citation redacted to preserve double blind anonymity, since the paper cites our de-anonymized paper)
> 		- Advancing natural language feedback-based approaches to alignment (due to presence of open-ended explanations for annotators’ preference choices)
>
> Additionally:
> 1. [L179-180: How do you check the quality of generated responses here?]
> - Good question. Our team (which included a social scientist with a political science background) manually curated and verified each set of responses. All responses are included in App H.
> 2. [Could the authors present Table 1 in a more concise format?]
> - Great idea; we have moved the table to the appendix and replaced it with a figure in the main paper. This gave us room to add an abridged related work :)
> 3. [Why did the authors not train or evaluate models using their collected dataset?]
> - We would like to point out that we have already run 96 experiments (including with RL, 70B model) to show that NC sampling alone (the methodology used to collect Community Alignment) already improves preference learning of all IW value dimensions. From there, rather than present the results of one specific way of using the new dataset, we instead emphasize the unique properties of CA from which many different analyses and experiments can emerge. Given the scope of the paper already (e.g., the multi-national & multilingual human survey (N=15,000) & paired model evaluation in Sec 2, almost 100 training experiments in Sec 3, massive data collection effort leading to the largest open-source preference dataset in Sec 4), we have chosen to leave the wide array of possible analyses and experiments to future work :) Indeed, we have spoken to many researchers who are currently utilizing Community Alignment (e.g. for social choice approaches to alignment, better understanding of cross-cultural preferences for LLMs, developing personalization methods, etc).
>
> Thank you for your questions and engagement with our paper. Please let us know if you have any outstanding concerns, and we would be happy to follow up.  And if your concerns have been addressed, we would really appreciate it if you would consider raising your score!

---

### Official Review · Reviewer_t1un · 2025-11-04

**Soundness:** 4
**Presentation:** 3
**Contribution:** 3
**Rating:** 6
**Confidence:** 4

**Summary:**

This paper reveals algorithmic monoculture in LLMs through a large-scale study showing that 21 state-of-the-art models produce responses aligned with only 41% of human preferences, particularly favoring secular-rational and self-expression values over traditional and survival-oriented values. The authors demonstrate that this homogeneity combined with temperature sampling prevents standard alignment methods (prompt-steering, SFT, DPO, GRPO) from learning diverse preferences from existing datasets. They propose negatively-correlated (NC) sampling to generate more diverse candidate responses and show that models trained on NC-sampled preference data achieve substantially better alignment with diverse values. Finally, they introduce Community Alignment, a new dataset of 200,000 multilingual preference comparisons built using NC sampling.

**Strengths:**

- Valuable dataset contribution: opensourcing a large-scale multilingual preference dataset with unique features (prompt-level annotator overlap   and comparison-level natural language explanations)
- Comprehensive and rigorous experiments and evaluation: Nationally representative samples from five countries, professional translations, and systematic evaluation of 21 LLMs. Testing four different approaches (prompt-steering, SFT, DPO, GRPO) shows the problem is fundamental rather than method-specific.
- Well motivated; Clear problem identification with practical solution: Effectively demonstrates that temperature sampling from 21 models fails to generate sufficient diversity. NC sampling is simple yet effective.

**Weaknesses:**

- Limited analysis of downstream performance trade-offs: No measurement of NC sampling's impact on general task performance or helpfulness. Prior work suggests diversity might degrade fine-grained learning signals and overall model quality. Without evaluating performance on standard benchmarks, practical applicability remains unclear
- Insufficient clarity in experimental procedures: Critical details are relegated to appendices or described too briefly. Section 3 lacks clear explanation of win rate computation, response generation from different model variants, and sampling parameters. The paper mentions a "GPT-4o-based judge" but doesn't explain the evaluation setup adequately
- Potential overstatement of results: The claim of "significant heterogeneity" in preferences (line 197) feels unsupported given Figure 1's concentrated distribution. Figure 2 mentions 20-40% coverage for temperature sampling but doesn't provide explicit percentages for NC sampling, making improvement magnitude unclear. Quantitative measures should support such claims. The paper occasionally uses imprecise language (e.g., "significant" without statistical tests)

**Questions:**

- What is the impact of NC sampling on standard NLP benchmarks (e.g., MMLU, general helpfulness)?
- What do the different lines in Figure 1's leftmost plot represent? Please add clearer legends.
- Figure 2 shows different models respond differently to NC sampling (Llama-3.3-70B minimal change, Claude-3.7-sonnet substantial). Do you have any insights on what accounts for these differences?
- How sensitive is NC sampling to the specific diversity prompt used? Have you tested variations?

---

> ### Author Response · Authors · 2025-11-21
> **Author response (part 1)**
>
> Thank you for recognizing the valuable dataset contribution, comprehensive and rigorous experiments and evaluation, and strong motivation for the work! Addressing your concerns under weaknesses below:
>
> 1. [Limited analysis of downstream performance trade-offs: No measurement of NC sampling's impact on general task performance or helpfulness. Prior work suggests diversity might degrade fine-grained learning signals and overall model quality. Without evaluating performance on standard benchmarks, practical applicability remains unclear]
>
> - While several datasets already exist for helpfulness or general task performance, there is no comparable large-scale preference dataset for pluralistic alignment–the goal of which is to adapt models to account for the diverse and heterogeneous preferences that different people have. Just as a math dataset would not be expected to improve alignment with varied values, our dataset was curated with the specific goal of advancing pluralistic alignment, rather than enhancing general task performance.
> - However, we recognize the reviewer’s concern that a more specialized dataset could potentially diminish general capabilities. In practice, though, this is typically addressed by combining multiple datasets (e.g., [1, 2]). For instance, one could train models using datasets for general helpfulness or task performance, while leveraging Community Alignment to tailor models to country-specific preferences.
> - In terms of practical applicability, we have spoken to several researchers who are now leveraging Community Alignment for the new use cases that it enables, as outlined in Sec 4.1 (e.g., social-choice-based approaches to alignment, cultural steerability, personalization, etc). Notably, one subsequent work has already used the Community Alignment dataset to create new interpretable personalization methods [3].
>
> [1] Unpacking DPO and PPO: Disentangling Best Practices for Learning from Preference Feedback. Ivison et al, NeurIPS 2024.
> [2] RLHF Workflow: From Reward Modeling to Online RLHF. Dong et al, TMLR 2024.
> [3] Citation redacted to preserve double blind anonymity (since the paper cites our de-anonymized paper)
>
> 2. [Insufficient clarity in experimental procedures: Critical details are relegated to appendices or described too briefly. Section 3 lacks clear explanation of win rate computation, response generation from different model variants, and sampling parameters. The paper mentions a "GPT-4o-based judge" but doesn't explain the evaluation setup adequately.]
> - Thank you for the comment. Given space constraints (appendix of 120 pages), we had to make a choice of which details to include in the main paper, but we have updated the manuscript to bring the specific details mentioned by the reviewer into the main paper (e.g., judge is a model prompted to end its response with its answer (0 or 1), win rate is computed on temperature one sampled generations).
> 3. [Potential overstatement of results: The claim of "significant heterogeneity" in preferences (line 197) feels unsupported given Figure 1's concentrated distribution. Figure 2 mentions 20-40% coverage for temperature sampling but doesn't provide explicit percentages for NC sampling, making improvement magnitude unclear. Quantitative measures should support such claims. The paper occasionally uses imprecise language (e.g., "significant" without statistical tests)]
> - We believe there might be a misunderstanding in some of the presented results. The concentrated distribution in the right plot of Figure 1 actually indicates the values encoded in LLM responses, whereas the left plot illustrates the heterogeneity of human preferences.
> - We have updated Figure 2 and added explicit percentages on NC sampling in the caption as well. In the course, we have also made a correction to the calculations in Fig 2 which shows that NC sampling yields even stronger benefits than previously reported. For temperature-sampling, the mean coverage of traditional and survival values, averaged across models, is 15% and 30%. With NC sampling, the mean coverage of traditional and survival is 60% and 53%. This means that NC sampling yields a 4x and 2x improvement in coverage for traditional and survival values, respectively, without any degradation, only improvements, in secular rational and self-expression coverage.
> - We used the term “significant” for  results that are clearly pronounced. For instance, for the claim that human preferences are significantly more heterogeneous than that of model responses, we see zero of the 420 model samples lie in the bottom half of the IW coordinates, whereas there is far from non-zero density in all quadrants in the human plot. For the claim that NC sampling significantly improves over temperature sampling, Table 1 presents error bars for all the win rate computations.
> - However, we understand the use of the term “significant” may be overloaded, and we have updated the paper to use alternate terms.

---

> > ### Author Response · Authors · 2025-11-21
> > **Author response (part 2)**
> >
> > Additionally, addressing your questions below:
> > 1. [What is the impact of NC sampling on standard NLP benchmarks (e.g., MMLU, general helpfulness)?]
> > - We believe this is outside the scope of this work which tackles pluralistic alignment in particular, but point the reviewer to our response to Weakness 1 and agree that future work on other NLP tasks could be interesting.
> > 2. [What do the different lines in Figure 1's leftmost plot represent? Please add clearer legends.]
> > - Could you please clarify what you mean? The only lines in Figure 1 are the dotted lines signifying the zero value (i.e., balanced) on the x and y axis, and the contour lines to represent the distribution of human preferences across the IW values.
> > 3. [Figure 2 shows different models respond differently to NC sampling (Llama-3.3-70B minimal change, Claude-3.7-sonnet substantial). Do you have any insights on what accounts for these differences?]
> > - Great question. We noticed that the difference is more pronounced in models whose default outputs are relatively homogenous. We hypothesize that these differences in baseline diversity may be related to the post-training applied to each model, though without access to the specific training recipes, it’s difficult to draw definitive conclusions.
> > 4. [How sensitive is NC sampling to the specific diversity prompt used? Have you tested variations?]
> > - Good question! We only tested one prompt (without extra prompt engineering) and found that it worked quite well out-of-the-box across a wide range of models. We imagine with the right optimization that one could potentially elicit even better results / more diversity.
> >
> > Thank you again for your thoughtful review. If you have any additional questions, please do not hesitate to ask!

---

### Author Response · Authors · 2025-11-21
**General response**

Thank you to all the reviewers for their thoughtful comments and feedback on our paper--we truly appreciate it.

We are excited to share that since the submission, we have collected additional data to improve demographic representativeness. The final dataset has a total of ~265k comparisons from >3600 annotators, with subsets of annotators from all five countries that are balanced across age, gender, and ethnicity.

As the largest and most demographically representative dataset of human preferences for LLM responses, Community Alignment unlocks many new directions for pluralistic alignment.  In fact, several researchers are already utilizing our Community Alignment dataset for work from social-choice-based approaches and cultural steering methods to personalization and more.

Thank you again for your feedback. We hope our responses have addressed your questions, but if you have any further concerns, please feel free to reach out—we would be happy to continue the discussion.

---

### Meta-Review · Area_Chair_cKQG · 2026-01-05

**Summary:**

While the paper is ambitious and well written, several reviewers raise substantive concerns that are particularly salient given that its core contribution is a dataset and data-collection methodology. Reviewer Y7NQ notes that, despite positioning Community Alignment as a flagship resource for pluralistic alignment, the paper provides limited dataset quality analysis and insufficient main-text documentation, which is especially problematic given the paper’s own framing of preference datasets as survey-like instruments sensitive to design choices. Relatedly, multiple reviewers (Y7NQ, rAhP) point out that the manuscript does not clearly contrast Community Alignment with prior preference datasets along the axes most relevant to its claims—annotator structure, demographic representativeness, language coverage, and downstream usability—making it difficult to assess what concrete shortcomings of existing datasets are resolved. The reliance on a narrow set of Inglehart–Welzel dimensions further constrains interpretability, a concern explicitly raised by Y7NQ, and the dataset’s practical value remains largely prospective, as no models are trained or evaluated directly using it. Finally, reviewers question whether negatively-correlated sampling represents a robust, general methodology or a prompt-level heuristic whose reliability may depend on model-specific behaviors. However, the high scores and generally positive attitude of the reviewers is sufficient to warrant acceptance.

**Reviewer Concerns:**

Several reviewers raise open questions that are particularly relevant given the paper’s dataset-centered contribution. Reviewer Y7NQ notes that the main text provides limited documentation and analysis of dataset quality and annotator behavior, and that more thorough characterization would help readers assess the reliability and scope of Community Alignment. Relatedly, reviewers suggest that the paper would benefit from a clearer, systematic comparison to prior preference datasets along key dimensions such as annotator structure, demographic and language coverage, and intended downstream use, in order to better situate its contributions. The reliance on a small set of Inglehart–Welzel dimensions also prompts questions about representativeness and construct validity, with reviewers recommending a more explicit discussion of the framework’s scope and limitations. In addition, while the dataset is positioned as enabling future pluralistic alignment work, reviewers note that its immediate practical utility would be clearer if accompanied by direct model training or evaluation using the dataset itself. Finally, there is interest in better understanding the generality of negatively-correlated sampling, including whether it constitutes a broadly applicable data-collection principle or a prompt-based technique whose effectiveness may vary across models and settings.

**Reviewer Scores:**

see above

---

### Decision · Program_Chairs · 2026-01-26

Accept (Poster)